# The Rice Serine/Arginine Splicing Factor RS33 Regulates Pre-mRNA Splicing during Abiotic Stress Responses

**DOI:** 10.3390/cells11111796

**Published:** 2022-05-30

**Authors:** Haroon Butt, Jeremie Bazin, Kasavajhala V. S. K. Prasad, Nourelislam Awad, Martin Crespi, Anireddy S. N. Reddy, Magdy M. Mahfouz

**Affiliations:** 1Laboratory for Genome Engineering and Synthetic Biology, Division of Biological and Environmental Sciences and Engineering, 4700 King Abdullah University of Science and Technology, Thuwal 23955-6900, Saudi Arabia; haroon.butt.1@kaust.edu.sa; 2CNRS, INRA, Institute of Plant Sciences Paris-Saclay IPS2, University Paris-Saclay and University of Paris Bâtiment 630, 91192 Gif sur Yvette, France; jeremie.bazin@ips2.universite-paris-saclay.fr (J.B.); martin.crespi@ips2.universite-paris-saclay.fr (M.C.); 3Department of Biology, Program in Cell and Molecular Biology, Colorado State University, Fort Collins, CO 80523, USA; kasavajhala.prasad@colostate.edu (K.V.S.K.P.); anireddy.reddy@colostate.edu (A.S.N.R.); 4Helmy Institute of Biomedical Science, Zewail City of Science and Technology, Ahmed Zewail Road, Giza 12578, Egypt; nawad@zewailcity.edu.eg

**Keywords:** pre-mRNA splicing, alternative splicing, SR proteins, genome engineering, abiotic stress

## Abstract

Abiotic stresses profoundly affect plant growth and development and limit crop productivity. Pre-mRNA splicing is a major form of gene regulation that helps plants cope with various stresses. Serine/arginine (SR)-rich splicing factors play a key role in pre-mRNA splicing to regulate different biological processes under stress conditions. Alternative splicing (AS) of SR transcripts and other transcripts of stress-responsive genes generates multiple splice isoforms that contribute to protein diversity, modulate gene expression, and affect plant stress tolerance. Here, we investigated the function of the plant-specific SR protein RS33 in regulating pre-mRNA splicing and abiotic stress responses in rice. The loss-of-function mutant *rs33* showed increased sensitivity to salt and low-temperature stresses. Genome-wide analyses of gene expression and splicing in wild-type and *rs33* seedlings subjected to these stresses identified multiple splice isoforms of stress-responsive genes whose AS are regulated by RS33. The number of RS33-regulated genes was much higher under low-temperature stress than under salt stress. Our results suggest that the plant-specific splicing factor RS33 plays a crucial role during plant responses to abiotic stresses.

## 1. Introduction

The splicing of pre-mRNAs, a key step in regulating gene expression, helps plants respond to stressful conditions such as salt, drought, and high or low temperatures [1,2,3,4,5,6,7,8]. The spliceosome is a megadalton protein complex that cleaves the introns, ligates the exons, and releases a mature mRNA. Chemical or genetic disruption of spliceosome function inhibits splicing in plants [9,10,11,12]. Spliceosome formation requires five small nuclear ribonucleoproteins (snRNPs: U1, U2, U4/U6, and U5) and ~200 additional proteins [13,14]. Splicing factors recognize the *cis*-regulatory elements of their target pre-mRNA and stabilize snRNP assembly across the introns and exons [14,15,16,17]. The splicing reaction starts with the binding of the U1 snRNP to the 5′ splice site and the U2 auxiliary factor (U2AF) to the 3′ splice site. This early complex (complex E) recruits the U2 snRNP to the branch-point (BP) sequence to form a pre-spliceosome complex (complex A). Complex A associates with the preassembled U5-U4/U6 tri-snRNP complex to form the first fully established spliceosome, known as the precatalytic spliceosome (complex B). Complex B is further converted into the activated spliceosome (complex B^act^) and catalytically activated spliceosome (complex B*), where the first transesterification reaction takes place. There are additional distinct, catalytically active spliceosome complexes, C and C*, which catalyze the second transesterification reaction that leads to the excision of the intervening intron and ligation of the adjoining exons [14,17,18].

Serine/arginine (SR)-rich proteins are conserved RNA-binding proteins that function as splicing factors and regulate pre-mRNA splicing by determining the splice sites [18,19,20,21,22,23,24]. SR proteins have one or two characteristic N-terminal RNA recognition motifs (RRMs) that determine their RNA-binding properties. The C-terminus of SR proteins is rich in alternating arginine and serine residues, referred to as the RS domain, and this domain facilitates protein–protein interactions [23,25,26,27]. The C-terminus of SR proteins can be reversibly phosphorylated, which affects the functions of SR proteins, including altering their splicing function [28,29]. In animal systems, the functions of SR proteins are not limited to pre-mRNA splicing and include mRNA nuclear export, mRNA stability, genome maintenance, and translation [19,20,30,31]. Plants have three distinct SR protein subfamilies that are not observed in the animal kingdom, suggesting that SR proteins likely have unique roles during plant growth and development [25,26].

Pre-mRNAs transcribed from multi-exon genes are either constitutively spliced or undergo alternative splicing (AS). AS produces different isoforms with alternative 5′ and 3′ splice sites, skipped exons, or retained introns. AS thus increases the plasticity of the genome by producing multiple isoforms from a single locus and thereby enhancing the protein diversity [1,4,32]. AS takes place when splicing factors select different splice sites in a closely regulated process [33,34,35]. Environmental stresses and developmental cues greatly influence the AS patterns of pre-mRNAs in plants [1,7,36,37,38]. The loss-of-function mutants of different SR family members, such as *rs40*, *rs41*, *scl30a*, and *sr45*, showed hypersensitivity to salt and abscisic acid (ABA) stress in *Arabidopsis thaliana* [39,40,41,42]. The pre-mRNAs of SR proteins are also extensively alternatively spliced [43,44], and splicing factor SR45 regulates the salt stress response in an isoform-dependent manner in Arabidopsis [45]. 

Rice (*Oryza sativa*) SR proteins also play important roles in mineral nutrient homeostasis, and SR40, SCL25, and SCL57 regulate inorganic phosphate (Pi) uptake and mobilization in shoots [46]. Similarly, *SR34b* expression increased under cadmium (Cd) stress, and the *sr34b* mutant showed increased Cd sensitivity [47]. Another SR-like protein, REDUCED RED-LIGHT RESPONSES IN CRY1CRY2 BACKGROUND1 (RRC1), together with SPLICING FACTOR FOR PHYTOCHROME SIGNALING (SFPS), interact with photoreceptor phytochrome B and regulates light-dependent pre-mRNA splicing to promote photomorphogenesis [48]. These studies show that SR proteins regulate AS in response to environmental stresses; however, the precise role of individual SR proteins and the splicing events they regulate are not clear.

The clustered regularly interspaced short palindromic repeat (CRISPR)/CRISPR-associated protein 9 (Cas9) adaptive immunity system of bacteria and archaea has been adapted for genome editing applications [49,50,51,52,53,54]. In plants, this molecular tool is utilized for trait engineering, directed evolution, and crop improvement [9,55,56,57,58,59]. Recently, we employed the CRISPR-Cas9 system to target all members of the *SR* gene family and produced single- and multigene knockout mutants in rice [24].

In this study, we investigated the function of RS33, a plant-specific splicing factor, in abiotic stress responses and AS in rice using a loss-of-function mutant. We found that the *rs33* mutation caused hypersensitivity to salt and low-temperature stresses. The RNA deep-sequencing data identified a subset of genes involved in salt and low-temperature stress responses, which are regulated by RS33. Our results provide a link between RS33-regulated AS and plant stress responses.

## 2. Materials and Methods

### 2.1. Plant Materials, Rice Transformation, and Genotyping of the Mutant Plants

*Oryza sativa* L. ssp. *japonica* cv. Nipponbare was used for all the experiments. The *rs33-1* mutant progeny was used for the phenotypic analysis [24]. For CRISPR/Cas9 editing, the expression of *Cas9* was driven by the *OsUBIQUITIN* promoter, and the sgRNAs were designed to target the 1st exon of the *RS33* locus *LOC_Os02g03040*. The sgRNA was synthesized as a polycistronic tRNA-gRNA (PTG) fragment with *Bsa*I overhangs for cloning under the *OsU3* promoter in pRGEB32.

*Agrobacterium tumefaciens*-mediated rice transformation was performed using strain EHA105, as described previously [24]. For genotyping, DNA was extracted from leaf samples, and PCR was conducted using gene-specific primers. Purified PCR products were cloned using the CloneJET PCR Cloning Kit (K1231). Sanger sequencing was used to analyze the mutations.

### 2.2. Phenotypic Analysis

Rice WT, *Cas9*, and homozygous mutant *rs33* seedlings were sown in a 2:1 mixture of ProMix BX and Green grade in 4-inch pots. The seedlings were allowed to grow until the seed-filling stage under controlled conditions in a growth chamber maintained at 60% relative humidity, temperatures of 27/25 °C, and a light intensity of 400 μmoles/m^2^/s, with a 16/8-h day/night photoperiod. Freshly harvested seeds of each genotype were used for carrying out stress tolerance assays. 

Salt sensitivity assays: To study the effect of salt on seed germination, seeds of individual genotypes were dehusked and surface-sterilized with 70% ethanol for 1 min, followed by a 2.5% bleach solution (with a few drops of Tween 20) for an additional 40 min. Subsequently, the seeds were thoroughly washed five times for 5 min each with sterile deionized water. The sterilized seeds were transferred to a Petri dish with sterile filter paper presoaked in ½-strength Murashige and Skoog (½ MS) medium supplemented with and without 100 mM NaCl, along with 0.5 g/L of MES. The Petri dishes were covered and incubated in a plant growth chamber maintained at 27 °C with 200 μmoles/m^2^/s of light under a 16/8-h day/night photoperiod. Seed germination was recorded manually every day for the stipulated time. The seeds were considered germinated after the emergence of the radicle. To test the effect of salt stress on seedling growth, the surface-sterilized seeds were placed on filter paper bridges in glass test tubes (20 cm long and 2.5 cm in diameter), each containing 10 mL of ½ MS medium supplemented with and without 100 mM NaCl. The test tubes with seeds were transferred to a growth chamber (under the above-mentioned conditions). The seedlings were allowed to be grown vertically. After the specified time, the seedlings were removed from the test tubes and dry blotted on filter paper, and the fresh weight and shoot length were measured. 

Low-temperature sensitivity assays: To test the low-temperature tolerance of these genotypes, surface-sterilized seeds were plated on ~70 mL of ½ MS medium supplemented with 0.5 g/L of MES and 0.8% Phytoblend in a square Petri dish. The Petri dishes were transferred to different growth chambers maintained at the indicated temperatures for a specified time. The Petri dishes were placed vertically to enable the equitable temperature treatment. Seed germination was recorded manually every day. The seeds were considered germinated after the emergence of the radicle.

No seed germination was noted at and below14 °C. Hence, the seeds that were initially subjected to the low-temperature treatments of 14 and 4 °C were subsequently transferred to 27 °C after 7 and 4 days, respectively, to enable seed germination. To test the effect of low temperature on seed germination and seedling growth, the surface-sterilized seeds were placed on filter paper bridges in glass test tubes, each containing 10 mL of ½ MS medium. The test tubes with seeds were transferred to a growth chamber maintained at the specified temperatures. The seeds were germinated, and the seedlings were allowed to be grown vertically. At the indicated times, the seedlings were removed from the test tubes and dry blotted on filter paper, and the fresh weight and shoot length were measured. 

### 2.3. RNA Isolation and RNA-Seq

Wild-type and mutant rice seeds were sterilized and transferred to ½ MS agar media (without sucrose) in glass jars for 7 days. For salt stress, 7-day-old seedlings were treated with 200 mM NaCl in ½ MS liquid media or mock-treated without NaCl for 3 and 6 h. For cold stress, 7-day-old seedlings were treated at 4 °C (in precooled ½ MS liquid media) for 3 and 6 h.

After the indicated treatments, total RNA was extracted from the seedlings using the Direct-zol RNA Miniprep Plus kit (Zymo Research, 17062 Murphy Ave., Irvine, CA 92614, USA). RNA samples with an RNA integrity number (RIN) ≥7.0 were selected for library construction and examined through a 2100 Bioanalyzer (Agilent Technologies, 5301 Stevens Creek Blvd. Santa Clara, CA 95051, USA). The TruSeq mRNA stranded kit was used for RNA seq library construction. The HiSeq-4000 platform is used to generate high-quality paired-end reads.

### 2.4. Analysis of RNA-Seq Data and Gene Functional Classification

RNA-seq data were analyzed as previously described [10].Briefly, Illumina adapters were removed from RNA-seq libraries, and read pairs were trimmed to the same length using Trimmomatic. TrimmomaticPE—LEADING:25, TRAILING:25, CROP:120, and MINLEN:120 arguments were used. Further alignment of the processed reads was done using STAR. The following processing pipeline was used: --outSAMtype BAM SortedByCoordinate --outReadsUnmapped Fastx --readFilesCommand zcat, --quantMode GeneCounts, --outFilterMultimapNmax 1. DEseq2 was used to estimate the significance of the differential gene expression, and FDR was used to correct the *p*-value during the pairwise comparison between genotypes. A gene was counted as differentially expressed if its adjusted *p*-value (FDR) was ≤0.01. Gene ontology enrichment analysis was done as described here [60] and simplified using GO slim terms.

rMATS v4.02 (Multivariate Analysis of Transcript Splicing (MATS): http://rnaseq-mats.sourceforge.net/; accessed on 24 June 2021) was used for differential splicing. The arguments used were: -nthread 4 --read-length 120-t paired --libType fr-firststrand. The gene annotation GTF file described here [46] was used as a reference annotation. Differentially spliced events were defined with FDR < 0.001 and |IncLevelDifference| > 0.

The UpSetR and eulerr R packages were used to compare the sets of DEGs or DAS. The PheatMap package was used for hierarchical clustering, and clusters were extracted using the cutree function. The log2 fold change (log2FC) values of the DEGs and IncLevelDifference of differentially retained introns were used to cluster DEGs and DAS. Lists of all comparisons of DEGs and DAS are listed in Appendix A, respectively. 

### 2.5. RT-PCR

DNA digestion of the total RNA was done using an RNase-Free DNase Set (Invitrogen Cat. No. 18068-015). SuperScript First-Strand Synthesis System (Invitrogen, 168 Third Avenue Waltham, MA 02451, USA) was used to generate cDNA from the DNA-digested total RNA. PCR conditions were: initial denaturation at 95 °C for 2 min, then 40 cycles of 95 °C for 30 s, annealing temperature according to the primers for 30 s, and 72 °C for a variable time (according to the amplicon length); then, there was a final elongation at 72 °C for 5 min. The primer sequences used for RT-PCR are listed in Appendix A.

## 3. Results

### 3.1. OsRS33 Regulates Gene Expression and Pre-mRNA Splicing of a Small Set of Genes

OsRS33, a plant-specific SR protein, contains two RRMs at the N-terminus and one RS domain at the C-terminus (Figure 1a). Using the CRISPR/Cas9 genome editing tool, we targeted the first exon of the *OsRS33* locus and recovered a knockout mutant with an 11-nt deletion and a 1-nt substitution (Figure 1a) [24]. The *rs33* mutant did not exhibit any obvious growth or developmental defects compared to the wild type (WT), except for a subtle reduction in seedling growth (Appendix A). 

To identify *RS33-*regulated genes, we performed a deep RNA sequencing analysis of seven-day-old WT and *rs33* seedlings. A differential gene expression analysis between the WT and *rs33* plants revealed a reduced expression of ~150 genes, while ~100 genes were upregulated (Figure 1b). To determine the role of *RS33* in regulating pre-mRNA splicing, we analyzed the genome-wide RNA deep sequencing data to identify altered splicing events in the *rs33* mutant compared with the WT. We identified ~600 differential splicing events between the WT and *rs33* and found a high frequency of intron retention (IR) events, followed by alternative 3′ splice sites (A3′S), alternative 5′ splice sites (A5′S), and exon skipping (SE) events (Figure 1c).

To further validate these differential splicing events, we analyzed the IR events of a set of genes whose splicing is regulated by *RS33* using reverse transcription PCR (RT-PCR). We observed the IR of *OsRS29* transcripts specifically in the *rs33* mutant as compared to the WT (Figure 1e). Similarly, the IR events for transcripts of putative pentatricopeptide genes with 949 bp were enriched in the *rs33* background (Figure 1e). A comparison of the differentially expressed genes (DEGs) with the differentially alternatively spliced (DAS) genes revealed that only eight genes were both differentially expressed and differentially spliced (Figure 1d). Our analysis did not show enrichment of any specific biological processes for the DEGs and DAS genes. Overall, our results indicate that *OsRS33* regulates the expression and splicing of a small set of genes during early seedling growth under the control conditions.

### 3.2. The Osrs33 Mutant Is Hypersensitive to Salt and Cold Stresses 

To characterize the function of *OsRS33* in abiotic stress responses in rice, we subjected WT, transgenic *Cas9* (vector control), and *rs33* plants to salt or low-temperature stress. To examine the effect of salt stress on seed germination and coleoptile greening, WT, *Cas9*, and *rs33* seeds were exposed to 100 mM NaCl for ten days. The germination rates of all three lines were affected by the salt stress. However, *rs33* seeds exhibited significantly enhanced sensitivity to salt stress compared to WT and *Cas9* seeds (indicated by a lower germination rate) (Figure 2a,b). Furthermore, the inhibition of coleoptile greening was also higher in *rs33* seeds (Figure 2a), and coleoptile emergence was significantly delayed in *rs33* seeds compared to WT seeds. Salt stress also significantly reduced the shoot growth and fresh weight of *rs33* seedlings compared to WT and *Cas9* plants (Figure 2c,d).

We also examined the WT, *Cas9*, and *rs33* phenotypes during seed germination under low-temperature stress. Based on our preliminary germination studies, we designed four temperature treatments (27 °C for 9 days (control condition), 20 °C for 9 days, 14 °C for seven days, followed by 27 °C for four days, and 4 °C for four days, followed by 27 °C for seven days).

At 27 °C, WT and *Cas9* seeds exhibited rapid germination within 3 days, while *rs33* mutants seeds exhibited slightly delayed germination (Figure 3a, top left panel), with 100% germination by day 4. At 20 °C, germination was significantly delayed in all three genotypes. For example, over 60% and 50% of WT and *Cas9* seeds, respectively, germinated by the end of the third day and reached 100% germination on the fifth day, whereas the *rs33* mutant showed only 40% germination on the fifth day and took 9 days to obtain 100% germination (Figure 3a, bottom left panel). When seeds were exposed to 14 °C for seven days and transferred to 27 °C for four days, both WT and *Cas9* seeds exhibited rapid germination (within 1 day) and reached 100% on the 8th day, while *rs33* seeds exhibited delayed germination by 3 days and reached 100% on the 11th day (Figure 3a, top right panel). After exposure to 4 °C for four days and transferring to 27 °C, both WT and *Cas9* seeds exhibited >90% germination within one day at 27 °C. However, the germination of *rs33* seeds was significantly delayed with a germination rate of 10% on day 7, 60% on day 8, and 80% on day 9 (Figure 3a, bottom right panel).

We further analyzed the effect of low temperature on seedling growth by estimating the shoot length and fresh weight. In general, irrespective of the genotype, exposure to the continuous low temperature of 20 °C caused a significant reduction in seedling growth (Figure 3b). However, the *rs33* seedlings exhibited a significant reduction in growth compared to WT and *Cas9* plants, indicating their enhanced sensitivity to low temperatures (Figure 3b). Taken together, our results indicate that the loss of RS33 significantly impacts the salt and cold tolerance of rice seedlings, highlighting the role of *OsRS33* in abiotic stress responses.

### 3.3. Loss of OsRS33 Leads to Genome-Wide Splicing Defects and Major Gene Expression Changes under Salt Stress

To further determine the function of *OsRS33* during salt stress at the gene expression level, 7-day-old *rs33* and WT seedlings were subjected to salt stress for 3 and 6 h, followed by an RNA-seq analysis. After 3 h of salt stress treatment, we identified 148 and 164 DEGs in *rs33* and WT plants, respectively (Figure 4a). However, after 6 h of salt stress treatment, there was a slight increase in the number of DEGs in the WT and a large increase in the number of DEGs (2344) in the *rs33* mutant (Figure 4a). A majority of these DEGs (2058) in the *rs33* mutant were unique and did not overlap with those in the WT (Appendix A). 

Hierarchical clustering revealed the expression profiles of the DEGs in response to salt stress, which were grouped into four co-expression modules (Figure 4b). Strikingly, the largest cluster (cluster 1) corresponded to a set of genes strongly associated with protein translation (p.adj < 1.6 × 10^−148^) and associated processes such as ribosome biogenesis (p.adj < 3.7 × 10^−121^). This cluster corresponded to genes that were repressed by salt stress in the *rs33* background as compared to the WT. By contrast, the genes of cluster 3 were specifically induced after 6 h of salt stress in the *rs33* mutant and were enriched for the genes associated with the response to water deprivation (p.adj < 2.41 × 10^−6^) and response to ABA (p.adj < 1.9 × 10^−4^) (Appendix A).

To further examine the global defects in pre-mRNA splicing, we used rMATS and defined AS events with FDR < 0.001 and |IncLevelDifference| > 0.2 as DAS. After comparing the control and salt stress conditions, we detected 1158 and 1116 nonredundant DAS events significantly regulated by salt stress in the WT and the *rs33* mutant, respectively. All types of AS events, such as IR, SE, A5′S, and A3′S events, were detected. Among these, differential IR represented the majority of events (Figure 4c). These AS events were mostly specific to the treatment time and the genotype and showed limited overlap (Figure 3a). Clustering of the DAS events in response to salt stress segregated them into five coregulated modules (Figure 4d and Appendix A). We then compared the DEGs with DAS events in *rs33* and WT plants after 3 h and 6 h of salt treatment (Figure 4e). Most of the DEGs were unique, and very few of them were alternatively spliced (Figure 4e), uncoupling the transcription from AS regulation. 

Moreover, we tested the IR of a few selected genes that were alternatively spliced in *rs33* compared to the WT after salt stress treatment by semi-quantitative RT-PCR using primers flanking the retained introns in these target genes. The pre-mRNA splicing factor SF2 was differentially spliced, and isoforms of 500 bp and 592 bp were enriched in the *rs33* mutant after salt treatment (Figure 4f, top left panel). For the helix-loop-helix DNA-binding domain-containing protein, the isoforms of 1807 bp and 3178 bp were differentially enriched between the WT and *rs33* plants (Figure 4f). The presence of both isoforms of *D-MANNOSE-BINDING LECTIN FAMILY PROTEIN* was significantly enhanced in the *rs33* mutant; similarly, the *UBIQUITIN CARBOXYL-TERMINAL HYDROLASE*, *FAMILY 1* isoform of 582 bp was increased in the *rs33* mutant (Figure 4f) under salt stress. These results indicate that RS33 regulates the splicing patterns of specific genes in response to salt stress.

### 3.4. Low Temperature Differentially Affects Global Gene Expression and AS in rs33 and WT Plants

In plants, cold stress impacts pre-mRNA splicing and gene expression. To investigate the role of *OsRS33* in cold-regulated gene expression we exposed 7-day-old *rs33* and WT seedlings to low-temperature (4 °C) stress for 3 h and 6 h, then performed RNA-seq and analyzed the data for DEGs and DAS events. Our analysis identified a large number of genes whose expression was regulated by low temperature. The number of DEGs was 1867 for WT-3h, 2153 for *rs33*-3h, 4972 for WT-6h, and 3777 for *rs33*-6h (Figure 5a). Interestingly, at the 3-h time point, *rs33* had a higher number of DEGs, while, at 6 h, the WT had a higher number of DEGs, indicating that the *rs33* mutant was less responsive to prolonged low-temperature stress. The unique DEGs numbered 1984 for WT-6h, 1230 for *rs33*-6h, 203 for WT-3h, and 347 for *rs33*-3h and showed substantial overlap (Figure 4a). 

Hierarchical clustering revealed five co-expressed modules, and their functional annotations were associated with photosynthesis and reactive oxygen species metabolic processes (cluster 1), amino acid metabolic process (cluster 2), xylan catabolic process and cell communication (cluster 3), glycoprotein metabolic process, cell wall organization and biogenesis, protein folding (cluster 4), phosphorus metabolic process, xylem development, and phosphorylation (cluster 5) (Figure 4b). Strikingly, cluster 5 showed the upregulation of phosphorylation and phosphorus metabolism genes at both time points in the *rs33* mutant, indicating a strong role of *RS33* in phosphorus metabolism during cold stress (Figure 4b and Figure 5b). Cluster 1 showed a strong upregulation of the oxidative stress response in *rs33* at 6 h of cold treatment as compared to the WT, while cluster 4 showed the downregulation of cell wall biogenesis in *rs33* at 6 h (Figure 4b and Figure 5b).

To further dissect the role of *RS33* during pre-mRNA splicing regulation under cold stress, we performed a splicing analysis and identified DAS events for each genotype and time point. All types of AS events were detected, with IR representing the majority of AS events (Figure 5c). Our analysis identified IR events numbering 1041 for WT-3h, 1091 for *rs33*-3h, 2148 for WT-6h, and 1868 for *rs33*-6h (Figure 5c). A large set of unique DAS events was identified for each genotype: 874 for WT-6h, 625 for *rs33*-6h, 370 for WT-3h, and 406 for *rs33*-3h (Figure 5a). 

Functional annotation of the individual DAS clusters revealed associations with regulation of the response to stress and lipid oxidation (cluster 1), signal transduction by protein phosphorylation, intracellular signal transduction, the disaccharide biosynthetic process (cluster 2), positive regulation of molecular function and activation of GTPase activity (cluster 3), histone ubiquitination and cellular ketone body metabolic process (cluster 4), and phospholipid metabolic process and dephosphorylation (cluster 5) (Figure 5b). Cluster 4 was upregulated in *rs33* after 3 h and 6 h of cold treatment compared to the WT, while cluster 5 was downregulated in *rs33* at 3 h and 6 h of cold treatment compared to the WT (Figure 5d).

We compared the DEGs with DAS events, and we found a big overlap between the DEGs and DAS in cold stress, indicating that many of the DEGs were also differentially spliced (Figure 5e). We validated the IR events for a set of genes via semi-quantitative RT-PCR and found an enrichment of introns in the *rs33* mutant seedlings after cold treatment (Figure 5f). We tested several genes with alternatively spliced pre-mRNAs, including those encoding WRKY1 (LOC_Os01g14440), bZIP transcription factor family protein (LOC_Os02g14910), RNA recognition motif-containing protein (LOC_Os03g17760), and caleosin-related protein/ABA-induced protein (LOC_Os02g50140), to test whether they exhibit intron retention. Our data showed that the *rs33* mutant seedlings had increased IR events during the pre-mRNA splicing of these genes in comparison to WT seedlings (Figure 5f). Overall, these results demonstrate that RS33 regulates cold stress responses at the posttranscriptional level and highlights a link between AS and cold tolerance. 

## 4. Discussion

Exploring adaptive strategies to withstand extreme environmental conditions is needed to develop new plant cultivars that can withstand climate change. Plants have evolved flexible and complex regulatory systems that allow them to adapt to abiotic stresses. AS is a complex, coordinated transcriptional regulatory mechanism, whereby multiple mRNA variants are produced from a single gene via differential intron removal and the retention of different exon combinations from pre-mRNAs [26,33,35]. AS is finely regulated by RNA-binding proteins that recognize sequence signals in RNA and regulate their splicing. SR proteins are RNA-binding proteins that contain one or two characteristic RRMs in the N-terminal portion and an RS domain at the C-terminal end. 

Here, we identified the role of a plant-specific SR protein, RS33, with two tandem RRMs, in which one (RRM2) is different from that of human ASF/SF2 [25,61]. Previous studies have shown that *RS33* is involved in zinc homeostasis [46]. Additionally, Isshiki et al. were unable to recover transgenic rice plants overexpressing *RS33* from repeated transformations, which further emphasized the essential role of this splicing factor during rice growth and development [43]. In our analysis of the *rs33* mutant, we identified only a small set of genes that were differentially expressed or differentially spliced under control conditions (Figure 1). Interestingly, *RS29* was alternatively spliced in the *rs33* mutant, suggesting that SR proteins regulate the splicing of other SR members [21,43,62,63]. We also observed IR in transcripts of *LOC_Os08g05750*, which encodes a putative pentatricopeptide (PPR) (Figure 1e). PPRs are targeted at chloroplasts and mitochondria and bind one or several organellar RNA transcripts and, thus, influence their expression [64,65,66]. A chloroplast retrograde signal was found to regulate the AS of the *RS33* homolog in Arabidopsis and is necessary for proper responses to varying light conditions [67]. RS33-mediated regulation of splicing of the chloroplast-localized PPR transcripts may be another level of chloroplast–nucleus communication.

The regulation of gene expression is important for plant survival and acclimation under abiotic stress conditions, and genome-wide studies have shown that pre-mRNA splicing is affected by stress treatments [7,8,68,69]. Splicing inhibition initiates abiotic stress signaling in plants [11,12], and the repression of splicing inhibition can overcome abiotic stress, resulting in stress tolerance [9,10]. The activities of spliceosome proteins and splicing factors are important for plant stress tolerance [1,44,70]. We found that the germination of *rs33* seeds and seedlings was much more sensitive to the 100 mM NaCl treatment than the WT plants (Figure 2). Gene expression analysis indicated that, after 6 h of salt treatment, a large number of genes were differentially expressed between the *rs33* mutant and the WT. A major set of these DEGs was related to translation and RNA biogenesis (cluster 1), which were downregulated in *rs33* plants after 6 h of treatment. Another major portion of DEGs was related to water deprivation and water transport (cluster 3), which were upregulated in *rs33* plants after 6 h of treatment, indicating that RS33 regulates water stress-related genes under salt stress conditions. However, the DAS genes that were highly expressed after 6 h of salt treatment were mainly related to the regulation of protein modification, histone ubiquitination (cluster 1), and DNA methylation (cluster 2). 

We observed a high number of DEGs (~2344) after only 6 h of salt treatment in the *rs33* mutant, but the number of DAS genes in the same treatment was similar to that in the WT. This large difference might be due to the aberrant splicing of some transcription factors in the *rs33* mutant after the salt treatment. We validated the aberrant splicing of some important targets, like pre-mRNA-splicing factor SF2, which has RNA-binding activity, and helix-loop-helix DNA-binding protein, which functions as a transcription factor and is characterized as *IRON-RELATED TRANSCRIPTION FACTOR 2* (*IRO2*). Iron homeostasis is essential for plant growth and development, and *IRO2* expression was elevated in response to glutamine in the regulation of the stress responses [71,72,73]. The phenotypic analysis indicated that the IRO2 overexpression lines germinated and grew better than the WT, while the IRO2-RNAi lines grew slower than the WT [74]. The D-mannose-binding lectin family protein/S-Domain receptor-like protein-11 (SDRLP-11) is characterized by Ser/Thr kinase domains and induced by salicylic acid [75,76]. Ubiquitin carboxyl-terminal hydrolase family 1/ubiquitin c-terminal hydrolase 5 (UCH5) is an uncharacterized protein. The UCH protein family functions in protein deubiquitination [77]. The Arabidopsis homologs *UCH1* and *UCH2* influence the auxin-dependent developmental pathways, while *UCH3* acts to maintain the period of the circadian clock at high temperatures, redundantly with *UCH1* and *UCH2* [78].

Additionally, we found a number of other genes involved in salt stress responses and are aberrantly spliced in the *rs33* mutant compared with WT under salt stress conditions. For example, *OsSTL1* (*SALT TOLERANCE LEVEL 1*; *LOC_Os04g02000*), a zinc finger family member, was identified as a candidate gene for salt tolerance in a genome-wide association study (GWAS) [79]. Another important protein, *NAC2* (*NAM-AFAT-CUC*; *LOC_Os04g38720*), a member of the NAC transcription factor that regulates many aspects of plant growth and development, was mis-spliced in the *rs33* mutant. The *OsNAC2* expression was enhanced in rice seedlings exposed to high-salt concentrations, and OsNAC2 accelerates salt-induced programmed cell death [80]. The *NAC2* gene is regulated by the microRNA miR164b. Interestingly, the transgenic plants overexpressing the miR164b-resistant form of *OsNAC2* had higher levels of drought and salt tolerance than the wild type [81]. Similarly, *OsNAC61*, which was aberrantly spliced in the *rs33* mutant, was upregulated after the salt stress treatment in elite rice cultivars [82]. The splicing of *OsGLYII-2* is also affected after the salt treatment in *rs33*. *OsGLYII-2* (*glutathione-responsive rice glyoxalase II-2*; *LOC_Os03g21460*) is the second enzyme of the glyoxalase pathway that detoxifies cytotoxic metabolite methylglyoxal (MG) and belongs to the superfamily of metallo-β-lactamases. *OsGLYII-2* is highly stress inducible and functions in salinity adaptation by maintaining a better photosynthesis efficiency [83,84].

We tested different low-temperature treatments (4 °C for four days, 14 °C for seven days, and 20 °C for nine days) and found that the *rs33* mutant was hypersensitive to these temperatures compared to the WT. We used seedlings treated at 4 °C for 3 h and 6 h for RNA sequencing. Our results revealed a high abundance of DEGs and DAS events in the WT as compared to the *rs33* mutant after cold stress treatment. However, a large number of these genes were unique and did not show overlap (Appendix A). Photosynthesis and reactive oxygen species metabolic genes were upregulated in *rs33*, while cell wall organization- and glycoprotein metabolic process-related genes were downregulated in *rs33* after 6 h of cold treatment. For splicing, a different set of genes was affected, as the genes responding to stress and involved in lipid oxidation were upregulated, whereas the genes related to protein phosphorylation and signal transduction were downregulated. 

We validated certain target genes showing significant changes during splicing. For example, *WRKY1* was alternatively spliced after the low-temperature treatment in the *rs33* mutant. The WRKYs are a family of transcription factors involved in various physiological processes, including biotic and abiotic stress responses and developmental processes. *OsWRKY62* and *OsWRKY76*, two genes of the WRKY IIa subfamily, undergo AS in response to pathogen infection [85,86]. The AS event we identified here would generate a truncated isoform of WRKY1. In Arabidopsis, it was found that the expression of WRKY1 was induced by salt stress, and the *wrky1* mutant conferred increased salt sensitivity [87].Another transcription factor, *bZIP19*, was alternatively spliced, which added 135 amino acids in-frame to produce a functional protein isoform. The bZIP19 and bZIP23 transcription factors are central regulators of the zinc deficiency response in Arabidopsis [88]. In Arabidopsis, *AtbZIP60*, an ortholog of *OsbZIP50*, was identified to be a target of *INOSITOL-REQUIRING ENZYME-1* (IRE1)-mediated splicing [89,90]. Overall, the loss of *rs33* generates more isoforms of the target genes as compared to WT. This might happen due to the lack of efficient splicing activity or aberrant splicing. In the absence of splicing factors, the spliceosome fails to distinguish the exon–intron junctions of the pre-mRNAs and produce multiple transcripts.

Recent developments in the genome-wide data analysis uncovered the roles of many spliceosome proteins that are important for plant stress tolerance. For example, *SUPERSENSITIVE TO ABSCISIC ACID AND DROUGHT 1 GENE* (*SAD1*), which encodes *SM-LIKE PROTEIN 5* (*LASM5*), the *sad1/lsm5* mutation, leads to increased sensitivity to drought stress and ABA [91,92]. Another mutant of the same family, *lsm4*, showed oversensitivity to drought and ABA in Arabidopsis [93]. The Arabidopsis protein U1A controls pre-mRNA splicing under salt stress, and the *atu1a* mutant is hypersensitive to salt stress [94]. An RNA-directed DNA methylation factor, RDM16, is important for splicing, and loss-of-function mutants of RDM16 are hypersensitive to salt and ABA [95]. Similar examples were found for low-temperature stress responses for mutants of spliceosomal proteins. For example, PRP31 regulates the formation of the U4/U6.U5 snRNP complex during splicing. The Arabidopsis *prp31* mutant is hypersensitive to cold stress [96]. Arabidopsis STABILIZED 1, similar to the human U5 snRNP, is necessary for low-temperature tolerance [97,98]. Recently, it was found that a DEAD-Box RNA Helicase protein, OsRH42, plays a critical role under cold stress to regulate pre-mRNA splicing in rice [6]. Our work on RS33 reinforces the link between spliceosomal proteins and stress tolerance and how these SR proteins function to help plants rapidly cope with changing environmental conditions. 

## 5. Conclusions

Our work revealed that *rs33* mutant seedlings are hypersensitive to abiotic stresses. The gene expression analysis showed that several genes are differentially spliced in *rs33* plants. Under normal conditions, the loss of *RS33* is irrelevant, possibly due to the existence of other SR proteins. By contrast, under stress conditions, like salt and low temperatures, RS33 is needed for the splicing of a large number of pre-mRNAs to acclimate to stress. Therefore, in the *rs33* mutant, the absence of RS33 resulted in aberrant pre-mRNA splicing of a large set of stress response genes and, subsequently, hypersensitivity to abiotic stress. The precise control of splicing factor homeostasis and manipulation of specific splicing machinery genes may be an effective approach to breed plants for tolerance to environmental stresses.

## Figures and Tables

**Figure 1 cells-11-01796-f001:**
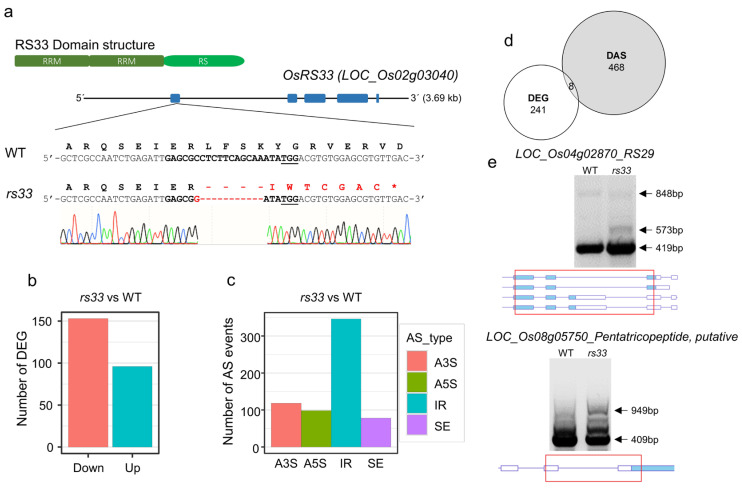
Genome-wide effects of the loss of *OsRS33* on alternative splicing in rice. (**a**) Targeted mutagenesis of the *OsRS33* (*LOC_Os02g03040*) locus in rice [24]. RS33 is a plant-specific SR protein characterized by the RNA Recognition Motif (RRM) and arginine–serine (RS) repeat domains. The *rs33* knockout mutant has an 11-nt deletion and a 1-nt substitution. (**b**) The number of up- and downregulated differentially expressed genes (DEGs) between the WT and *rs33*. (**c**) Bar plot showing the number of each type of differential alternative splicing (DAS) event mediated by RS33; the majority of these AS events were intron retention (IR) events. SE, exon skipping; A5′S, 5′ alternative splice site; A3′S, 3′ alternative splice site. (**d**) Venn diagram displaying the overlap between DEGs and DAS genes. (**e**) cDNA was prepared from one-week-old WT and *rs33* seedlings. RT-PCR was performed using primers that flank introns subject to AS in selected genes. Intron retention was observed in *rs33* mutant rice plants. Arrowheads indicate splicing variants. The gene structures and retained introns are shown. Red boxes indicate the PCR fragments.

**Figure 2 cells-11-01796-f002:**
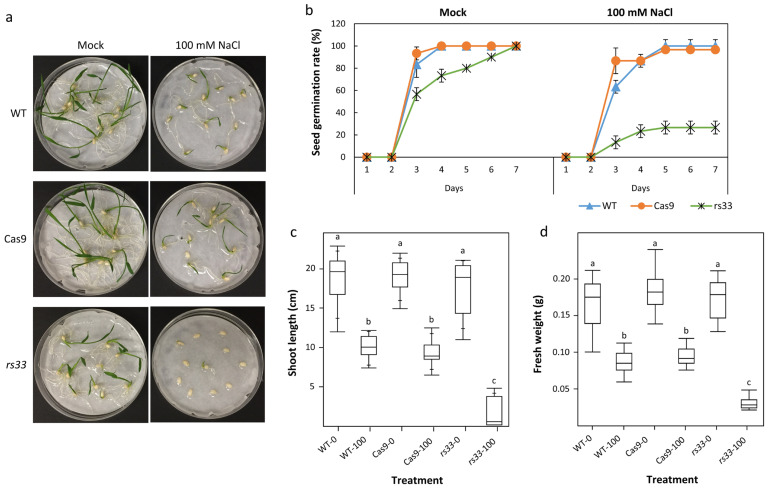
The *rs33* mutant is hypersensitive to salt stress. (**a**,**b**) Seeds of *rs33* and WT were germinated for ten days in ½ MS media supplemented with 0 mM and 100 mM NaCl. Germination was severely inhibited in *rs33* compared to WT seeds. Other growth parameters—shoot length (**c**) and seedling fresh weight (**d**)—were also severely affected in *rs33* as compared to WT seedlings. In c, d; significance is represented by letters. All the variables with the same letters are not statistically significant. If two variables have different letters, they are significantly different.

**Figure 3 cells-11-01796-f003:**
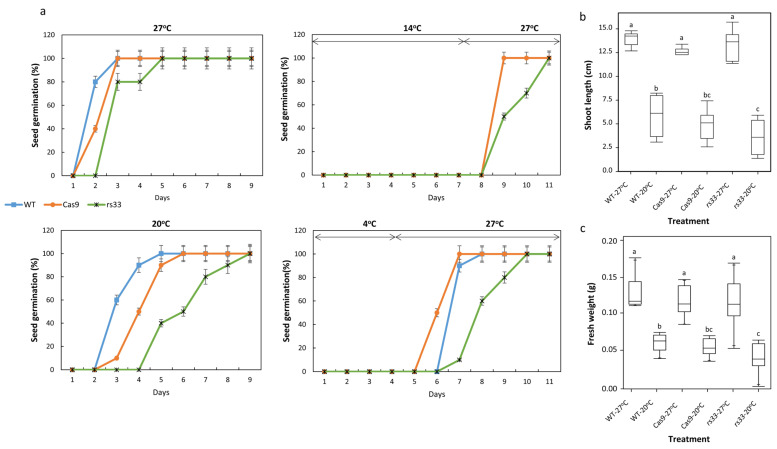
The *rs33* mutant is hypersensitive to cold stress. (**a**) The *rs33* mutant and WT seeds were germinated for ten days at the control temperature (27 °C) or different low temperatures (20 °C, 14 °C, and 4 °C). For 14 °C, the seeds were first kept at 14 °C for seven days and then transferred to 27 °C for 4 days. For 4 °C, the seeds were initially treated at 4 °C for four days and then germinated at 27 °C for seven days. At 20 °C, the germination of the *rs33* seeds was significantly delayed compared to that of the WT. The shoot length (**b**) and seedling fresh weight (**c**) were significantly affected in *rs33* as compared to WT seedlings. In b, c; all the variables with the same letters are not statistically significant. If two variables have different letters, they are significantly different.

**Figure 4 cells-11-01796-f004:**
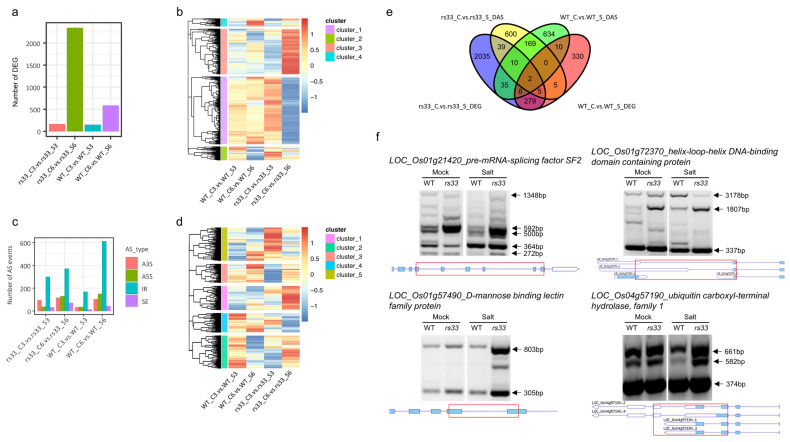
Genome-wide effects of the loss of *OsRS33* on gene expression and RNA splicing under salt stress. (**a**) Number of differentially expressed genes (DEGs) between the *rs33* mutant and the WT. One-week-old rice seedlings were treated with 200 mM NaCl for 3 and 6 h and used for RNA extraction. A high number of DEGs was observed for *rs33* seedlings after a 6-h treatment with NaCl. (**b**) Clustering analysis of the DEGs. Cluster 1 clearly shows the downregulation of translation/ribosome biogenesis genes in *rs33* in the 6-h NaCl treatment. Cluster 3 shows the upregulation of stress-responsive genes in *rs33* in the 6-h salt treatment as compared to the WT. (**c**) Bar plot showing the number of each type of differential alternative splicing (DAS) event induced by salt treatment in the *rs33* mutant and the WT. (**d**) Heatmap and hierarchical clustering of DAS genes after salt treatment in *rs33* and WT. (**e**) Venn diagram displaying the overlap between DEGs and DAS genes. (**f**) cDNA was prepared from one-week-old WT and *rs33* seedlings treated with 200 mM NaCl for 3 and 6 h. Mock-treated samples were used as controls. RT-PCR was performed using primers that flank introns subject to AS in the selected genes. Arrowheads indicate splicing variants. The gene structures and retained introns are shown. Red boxes indicate the PCR fragments. C3; control 3 h, C6; control 6 h, S3; salt 3 h, and S6; salt 6 h.

**Figure 5 cells-11-01796-f005:**
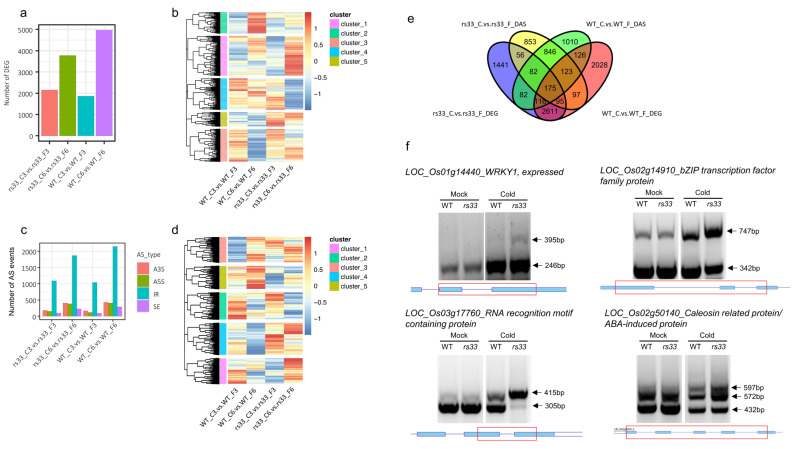
Genome-wide effects of the loss of OsRS33 on gene expression and RNA splicing under low-temperature stress. (**a**) A number of differentially expressed genes (DEGs) between the *rs33* mutant and the WT. One-week-old rice seedlings were treated with low-temperature stress (4 °C) for 3 and 6 h and used for RNA extraction. (**b**) Clustering analysis of the DEGs. Cluster 1 shows the upregulation of oxidative stress response genes in *rs33* after 6 h of cold treatment as compared to the WT. Cluster 4 shows that the cell wall biogenesis genes were downregulated after 6 h of cold treatment in *rs33*. (**c**) Bar plot showing the number of each type of differential alternative splicing (DAS) event induced by low-temperature stress in the *rs33* mutant and the WT. (**d**) Heatmap and hierarchical clustering of DAS genes after the cold treatment in *rs33* and WT. Cluster 1 contains stress-related genes with quick AS regulation upon stress. Cluster 5 indicates that some genes of the lipid metabolism pathways do not respond to cold in *rs33*. (**e**) Venn diagram displaying the overlap between DEGs and DAS genes. (**f**) cDNA was prepared from one-week-old WT and *rs33* seedlings treated at 4 °C for 3 and 6 h. Mock-treated samples were used as the controls. RT-PCR was performed using primers that flank introns subject to AS in the selected genes. Arrowheads indicate splicing variants. The gene structures and retained introns are shown. Red boxes indicate the PCR fragments. C3; control 3 h, C6; control 6 h, F3; 4 °C 3 h, and F6; 4 °C 6 h.

## Data Availability

The RAW data and processed data files were deposited in GEO (GEO; https://www.ncbi.nlm.nih.gov/geo/; accessed on 24 January 2022) under accession number GSE194283. The following secure token was created to allow a review of the record GSE194283, while it remains in a private status: etopyiycdhcllgb. Please note the following points: This token allows anonymous, read-only access to GSE194283 and its associated accessions while they are private. Treat the token as you would a password and realize that the token provides access to GSE194283 to anyone who uses it.

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
