# Peer review of "The Rice Serine/Arginine Splicing Factor RS33 Regulates Pre-mRNA Splicing during Abiotic Stress Responses"

_cells, 2022, doi:10.3390/cells11111796_

Round 1
Reviewer 1 Report
-
This manuscript addresses an important function of rice RS33 in response to abiotic stresses. The authors propose that the loss of function of RS33 changes the gene expression and splicing level in control, salt stress and cold stress. The results validate RS33 have potential roles involved in dealing with environmental stresses in rice. However, here are some concerns need to be addressed:
- It would be interesting and necessary to see/provide whether the phytohormone ABA concentration and/or response to ABA were affected in the sr33 mutant, both under control and salt/cold stress.
- Exon Skipping is denoted by “ES”, please check if the abbreviation is wrong.
- In section 3, Fig.4e shows limited overlap between DEGs and DAS genes, whether it indicates the functions of RS33 protein are not limited to pre-mRNA splicing。
- In section 3, pre-mRNA splicing factor SF2 is spliced in rs33. Is SF2 involved in alternative splicing induced by stress?
- In section 3, SR proteins as splicing factors generates produces different isoforms, the authors need to explain why the loss of RS33 produces more transcripts.
- In section 4, splicing target genes of RS33 in discussion have little connection to salt stress. The authors may discuss more on the relationship between targets and salt stress.
Author Response
This manuscript addresses an important function of rice RS33 in response to abiotic stresses. The authors propose that the loss of function of RS33 changes the gene expression and splicing level in control, salt stress and cold stress. The results validate RS33 have potential roles involved in dealing with environmental stresses in rice. However, here are some concerns need to be addressed:
Response: We thank the reviewer for raising these important questions. We have addressed all the concerns. Our detailed response to each comment is provided below.
1- It would be interesting and necessary to see/provide whether the phytohormone ABA concentration and/or response to ABA were affected in the rs33 mutant, both under control and salt/cold stress.
Response: We thank the reviewer for raising this important point. We agree with the reviewer that phytohormone ABA plays important roles in pant development and responses to abiotic stresses. The pre-mRNA splicing regulates the ABA signaling components like HAB1, PYL7, PYL8, ABI1, and ABI5 and the SR proteins coordinate plant abiotic stress responses by targeting ABA pathway components. The abiotic stresses including salt stress induce the expression of ABA-responsive genes and ABA accumulation revealing that ABA plays an essential role in salt-modulated growth. In our study we used salt stress to identify multiple splice isoform of stress-responsive genes, which are regulated by RS33. The role of SR proteins during the biogenesis of the phytohormone ABA and molecular mechanisms underlying the ABA-regulated plant growth are important parameters to study. However, we believe that these questions are beyond the scope of this study.
2- Exon Skipping is denoted by “ES”, please check if the abbreviation is wrong.
Response: We thank the reviewer for this comment. The abbreviation ‘ES’ and ‘SE’ are alternatively used for Exon Skipping/Skipped Exons.
3- In section 3, Fig.4e shows limited overlap between DEGs and DAS genes, whether it indicates the functions of RS33 protein are not limited to pre-mRNA splicing.
Response: We thank the reviewer for this excellent comment. We agree with the reviewer that DEGs are not overlapping with DAS genes after salt stress treatments. We described it in the manuscript L355-L357, and discussed at L489-L492. The effect of SR33 on DEGs might be indirect due to aberrant splicing of transcription factors, which in turn regulate the transcription.
4- In section 3, pre-mRNA splicing factor SF2 is spliced in rs33. Is SF2 involved in alternative splicing induced by stress?
Response: We thank the reviewer for this comment. The SF2 is not characterized in rice, however, its ortholog in Arabidopsis AtSR34 (AT1G02840) is involved in ABA-mediated stress responses.
Cruz, T.M.D.; Carvalho, R.F.; Richardson, D.N.; Duque, P. Abscisic Acid (ABA) Regulation of Arabidopsis SR Protein Gene Expression. Int. J. Mol. Sci. 2014, 15, 17541-17564. https://doi.org/10.3390/ijms151017541
5- In section 3, SR proteins as splicing factors generates/produces different isoforms, the authors need to explain why the loss of RS33 produces more transcripts.
Response: We thank the reviewer for this excellent comment. We added the following sentences in the revised version of the manuscript.
“Overall, the loss of rs33 generates more isoforms of the target genes compared to WT. This might happen due to the lack of efficient splicing activity or aberrant splicing. In the absence of splicing factors, the spliceosome fails to distinguish the exon-intron junctions of the pre-mRNAs and produce multiple transcripts”.
6- In section 4, splicing target genes of RS33 in discussion have little connection to salt stress.
Response: We thank the reviewer for this comment. We partially agree with the reviewer. The SF2 is a splicing factor although not characterized in rice but its ortholog in Arabidopsis AtSR34 (AT1G02840) is induced by stress.
Cruz, T.M.D.; Carvalho, R.F.; Richardson, D.N.; Duque, P. Abscisic Acid (ABA) Regulation of Arabidopsis SR Protein Gene Expression. Int. J. Mol. Sci. 2014, 15, 17541-17564. https://doi.org/10.3390/ijms151017541
The IRO2 is a transcription factor, which is involved in the regulation of stress responses in response to glutamine.
Kan, CC., Chung, TY., Juo, YA. et al. Glutamine rapidly induces the expression of key transcription factor genes involved in nitrogen and stress responses in rice roots. BMC Genomics 16, 731 (2015). https://doi.org/10.1186/s12864-015-1892-7
Ogo, Y., Itai, R.N., Kobayashi, T. et al. OsIRO2 is responsible for iron utilization in rice and improves growth and yield in calcareous soil. Plant Mol Biol 75, 593–605 (2011). https://doi.org/10.1007/s11103-011-9752-6
7- The authors may discuss more on the relationship between targets and salt stress.
Response: We thank the reviewer for this comment. We added the following paragraph in the discussion part in the revised version of the manuscript:
“Additionally, we found a number of other genes involved in salt stress responses and are aberrantly spliced in rs33 mutant compared with WT under salt stress conditions. For example, OsSTL1 (SALT TOLERANCE LEVEL 1; LOC_Os04g02000), a zinc finger family member, was identified as a candidate gene for salt tolerance in a genome-wide association study (GWAS) (Yuan et al., 2020). Another important protein NAC2 (NAM-AFAT-CUC; LOC_Os04g38720), a member of NAC transcription factor that regulate many aspects of plant growth and development, was mis-spliced in rs33 mutant. The OsNAC2 expression was enhanced in rice seedlings exposed to a high salt concentrations and OsNAC2 accelerates salt- induced programmed cell death (Mao et al., 2018). The NAC2 gene is regulated by the microRNA miR164b. Interestingly, the transgenic plants overexpressing the miR164b-resistant form of OsNAC2 had higher levels of drought and salt tolerance than the wild type (Jiang et al., 2019). Similarly, OsNAC61, which was aberrantly spliced in rs33 mutant, was upregulated after salt stress treatment in elite rice cultivars (García-Morales et al., 2014). The splicing of OsGLYII-2 is also affected after salt treatment in rs33. The OsGLYII-2 (glutathione responsive rice glyoxalase II-2; LOC_Os03g21460), the second enzyme of glyoxalase pathway that detoxifies cytotoxic metabolite methylglyoxal (MG), belongs to the superfamily of metallo-β-lactamases. The OsGLYII-2 is highly stress inducible and functions in salinity adaptation by maintaining better photosynthesis efficiency (Mustafiz et al., 2011; Ghosh et al., 2014).”
Reviewer 2 Report
Dear Authors,
The article entitled “The rice serine/arginine splicing factor RS33 regulates pre-mRNA
splicing during abiotic stress responses” describe alternative splicing genes altered into splicing under different abiotic stress in rice. The article Is well written, the results are rightly described, and the bibliography is updated and adequate. However, the quantity of the results obtained and shown is short based on the big data analysis performed.
To improve the quality of the work I would suggest:
1.- Comparing your RNAseq data with another analysis of alternative splicing sequencing to determine if the degree of alternative splicing changes produced under your conditions at the mutant compared to the wild type induce enough changes to determine that this gene is specifically responsible for the alternative splicing that you suggest.
2.- To get a direct relation between R33 and alternative splicing I would suggest also add new assays demonstrating a direct role of this gene (RS33), it is necessary to demonstrate this role in a wider way, not only in a few genes.
3.- The evidence shown at this work are indirect, so some assays of pull down might be useful to determine this direct role on salt and cold stress in rice.
Hoping to be useful
Author Response
The article entitled “The rice serine/arginine splicing factor RS33 regulates pre-mRNA splicing during abiotic stress responses” describe alternative splicing genes altered into splicing under different abiotic stress in rice. The article is well written, the results are rightly described, and the bibliography is updated and adequate. However, the quantity of the results obtained and shown is short based on the big data analysis performed.
To improve the quality of the work I would suggest:
1- Comparing your RNAseq data with another analysis of alternative splicing sequencing to determine if the degree of alternative splicing changes produced under your conditions at the mutant compared to the wild type induce enough changes to determine that this gene is specifically responsible for the alternative splicing that you suggest.
2- To get a direct relation between R33 and alternative splicing I would suggest also add new assays demonstrating a direct role of this gene (RS33), it is necessary to demonstrate this role in a wider way, not only in a few genes.
3- The evidence shown at this work are indirect, so some assays of pull down might be useful to determine this direct role on salt and cold stress in rice.
Response: We thank the reviewer for raising these questions. However, we feel that it is not necessary to reanalyze our RNA-seq for AS detection with another tool for the following reasons. First, we have used a very robust and widely used computational tool called rMATS that was published in PNAS (rMATS v4.02 (http://rnaseq-mats.sourceforge.net/;). Many papers that used only this tool were published (including our own work that was published in Communications Biology last year) in top-tier journals. Furthermore, we have validated AS of randomly selected genes using semi-quantitative RT-PCR. Hence, we are confident that our AS analysis results represent true AS events. Furthermore, our comprehensive phenotypic analysis provided convincing evidence that the rs33 mutant is overly-sensitive to abiotic stresses like salt and cold. This conclusion is validated further through the RNA-seq and molecular analysis. Comparison of AS results with another RNA-seq analysis pipeline, designing of new assays, and pull-down may be nice experiments for future dissection of the molecular mechanism involved in RS33 action. These suggested analyses/experiments are beyond the scope of this manuscript and are not necessary to support the conclusions of this manuscript.
Reviewer 3 Report
cells-1665040
The rice serine/arginine splicing factor RS33 regulates pre-2 mRNA splicing during abiotic stress responses
Haroon Butt, Jeremie Bazin, Kasavajhala V.S.K. Prasad, Nour El-Islam, Martin Crespi, Anireddy S.N. Reddy and Magdy M. Mahfouz
The authors investigated the function of the plant-specific SR protein RS33 in regulating pre-mRNA splicing under abiotic stress responses in rice. They showed that loss-of-function mutant rs33 showed increased sensitivity to salt and low-temperature stresses. Genome-wide analyses of gene expression and splicing in rs33 seedlings subjected to these stresses identified multiple splice isoforms of stress-responsive genes. The number of RS33-regulated genes was much higher under low-temperature stress than under salt stress. They conclude that the plant-specific splicing factor RS33 plays crucial roles during plant responses to abiotic stresses.
The results in the manuscript are potentially interesting. However, the authors need to address my concerns described below.
1) In RS33 KO plant, the authors did not show evidence for loss of RS33 protein. The authors have to demonstrate it by western blotting, at least by RT-PCR.
2) In Figure 1e, the 945bp band for Glutamate synthase is too faint to be convinced.
3) The authors successfully identified several genes whose ASs are changed in rs33 mutant under stress conditions. It would be nicer if the authors could explain the phenotypes with some of those genes in Discussion section.
Author Response
The authors investigated the function of the plant-specific SR protein RS33 in regulating pre- mRNA splicing under abiotic stress responses in rice. They showed that loss-of-function mutant rs33 showed increased sensitivity to salt and low-temperature stresses. Genome-wide analyses of gene expression and splicing in rs33 seedlings subjected to these stresses identified multiple splice isoforms of stress-responsive genes. The number of RS33-regulated genes was much higher under low-temperature stress than under salt stress. They conclude that the plant- specific splicing factor RS33 plays crucial roles during plant responses to abiotic stresses.
The results in the manuscript are potentially interesting. However, the authors need to address my concerns described below.
Response: We thank the reviewer for raising these important questions. Below, we have addressed all the points in detail.
1- In RS33 KO plant, the authors did not show evidence for loss of RS33 protein. The authors have to demonstrate it by western blotting, at least by RT-PCR.
Response: We thank the reviewer for this comment. We developed this k/o mutant via CRISPR/Cas9 genome engineering. We confirmed the mutation in the genome via Sanger sequencing (Figure 1). The Western blotting and RT-PCR analysis are needed for knock-down mutants where the protein may remain. However, in our case, we believe that Western blotting and RT-PCR analysis are not needed since the mutation leads to a null allele and has been confirmed in the genome through sequencing.
2- In Figure 1e, the 945bp band for Glutamate synthase is too faint to be convinced.
Response: We thank the reviewer for this comment. We agree with reviewer that this band is faint but we repeated the experiments twice and got the similar results. However, for the sake of smooth reading of the manuscript we have removed the Glutamate synthase from the Figure-1 and from the results and discussion part, in the revised version of the manuscript.
3- The authors successfully identified several genes whose ASs are changed in rs33 mutant under stress conditions. It would be nicer if the authors could explain the phenotypes with some of those genes in Discussion section.
Response: We thank the reviewer for this excellent comment. We added the following paragraphs in the discussion part in the revised version of the manuscript.
“The phenotypic analysis indicate that IRO2-overexpression lines germinate and grow better than WT, while IRO2-RNAi lines grow slower than WT (Ogo et al., 2011).”
“In Arabidopsis, it was found that the expression of WRKY1 was induced by salt stress, and wrky1 mutant confers increased salt sensitivity (Wu et al., 2022).”
“Additionally, we found a number of other genes involved in salt stress responses and are aberrantly spliced in rs33 mutant compared with WT under salt stress conditions. For example, OsSTL1 (SALT TOLERANCE LEVEL 1; LOC_Os04g02000), a zinc finger family member, was identified as a candidate gene for salt tolerance in a genome-wide association study (GWAS) (Yuan et al., 2020). Another important protein NAC2 (NAM-AFAT-CUC; LOC_Os04g38720), a member of NAC transcription factor that regulates many aspects of plant growth and development, was mis-spliced in rs33 mutant. The OsNAC2 expression was enhanced in rice seedlings exposed to a high salt concentrations and OsNAC2 accelerates salt- induced programmed cell death (Mao et al., 2018). The NAC2 gene is regulated by the microRNA miR164b. Interestingly, the transgenic plants overexpressing the miR164b-resistant form of OsNAC2 had higher levels of drought and salt tolerance than the wild type (Jiang et al., 2019). Similarly, OsNAC61, which was aberrantly spliced in rs33 mutant, was upregulated after salt stress treatment in elite rice cultivars (García-Morales et al., 2014). The splicing of OsGLYII-2 is also affected after salt treatment in rs33. The OsGLYII-2 (glutathione responsive rice glyoxalase II-2; LOC_Os03g21460) is the second enzyme of glyoxalase pathway that detoxifies cytotoxic metabolite methylglyoxal (MG), belongs to the superfamily of metallo-β-lactamases. The OsGLYII-2 is highly stress inducible and functions in salinity adaptation by maintaining better photosynthesis efficiency (Mustafiz et al., 2011; Ghosh et al., 2014).”
Round 2
Reviewer 1 Report
The revised manuscript is fine with me. I do not have any concern.
Reviewer 3 Report
The authors addressed most of my concerns. Although I still think RT-PCR and/or western blotting is required to confirm knockout of genes, I leave this issue for the editor's decision.